# Can the GRE predict valued outcomes? Dropout and writing skill

**Brent Bridgeman**[1]*, **Frederick Cline**[2]

**1** Educational Testing Service, Princeton, New Jersey, United States of America, **2** Retired, Lawrence Township, New Jersey, United States of America

* bbridgeman@ets.org

**Data Availability Statement:** Data has been deposited at openicpsr (https://www.openicpsr.org/openicpsr/). Dataset number is: 155721.

**Funding:** This research was funded by Educational Testing Service (BB and FC); there is no grant number. Encouragement to pursue this research

## Abstract

Graduate school programs that are considering dropping the GRE as an admissions tool often focus on claims that the test is biased and does not predict valued outcomes. This paper addresses the bias issue and provides evidence related to the prediction of valued outcomes. Two studies are included. The first study used data from chemistry (N = 315) and computer engineering (N = 389) programs from a flagship state university and an Ivy League university to demonstrate the ability of the GRE to predict dropout. Dropout prediction for the chemistry programs was both statistically and practically significant for the GRE quantitative (GRE-Q) scores, but not for the verbal or analytical writing scores. In the computer engineering programs, significant dropout prediction by GRE-Q was evident only for domestic students. In the second study, GRE Analytical Writing scores for 217 students were related to writing produced as part of graduate school coursework and relationships were noted that were both practically and statistically significant.

## Introduction

A number of graduate programs that formerly required all applicants to submit scores on the GRE General Test (GRE) have recently dropped that requirement [1]. Two of the most commonly cited reasons for this change are that the GRE is biased and that it does not predict the outcomes that graduate faculty value most, especially completion of PhD programs [2, 3].

The bias claim is based on an inappropriate equating of bias with the demonstration of group differences. In general, measurement instruments that reveal group differences are not said to be biased. A tape measure that shows that adult men are on average taller than adult women is not said to exhibit gender bias. And a thermometer that shows that people with the flu typically have a higher temperature than healthy people is not said to be biased against people with the flu. But an educational test that shows that students who, on average, had fewer educational opportunities and poorer schools get lower scores than more privileged students is sometimes said to be biased. This definition of bias has serious unintended consequences as it suggests that merely doing away with the test will address the underlying societal problem that the test reveals. But dropping the test will be as effective in more fairly allocating educational resources as destroying thermometers would be in combating the flu. This is not to say that

came from the graduate deans on the GRE Board. Nothing in this report represents an official position of the deans or Educational Testing Service. The funders had no role in study design, data collection and analysis, decision to publish, or preparation of the manuscript.

**Competing interests:** I have read the journal's policy and the authors of this manuscript have the following competing interests: Financial support for this research was provided by Educational Testing Service which encourages the independence of researchers; nothing in this report represents an official position of ETS".

biased test questions cannot exist, but that merely showing a group difference is not evidence of bias in the test. Bias and fairness training for test question writers and statistical tools for uncovering biased questions are still essential in the creation of fair and valid tests [4]. Trained fairness reviewers, including representatives of minority groups, review all GRE questions and statistical differential item functioning (DIF) procedures are used to identify any test questions that are unusually difficult (or easy) for a particular racial/ethnic or gender group.

The second reason cited for dropping a GRE requirement is that the scores do not predict valued outcomes. Although there is ample evidence that the scores can predict graduate school readiness as indexed by grades in the first one or two years (e.g., [5, 6]) mere grade prediction is of limited value. Faculty would really like to know which applicants are likely to complete their programs and which applicants demonstrate evidence of research skills (e.g., [2, 3]).

A number of studies have explored the extent to which GRE scores can predict research publications or program completion. One study was entitled, "The limitations of the GRE in predicting success in biomedical graduate school" [2]. GRE scores were used as one criterion for admission to this program, and the authors noted that GRE Quantitative (GRE-Q) scores were essentially uncorrelated with first author publication count. Another study was entitled, "Multi-institutional study of GRE scores as predictors of STEM PhD degree completion: GRE gets a low mark" [3]. The abstract of this study noted, "Remarkably, GRE scores were significantly higher for men who left than counterparts who completed STEM PhD degrees." A study of first author publications conducted in the biomedical program at the University of North Carolina-Chapel Hill [7] noted that GRE-Q scores are uncorrelated with first author publications.

Initially, these studies may appear to provide a compelling case against the use of GRE scores in graduate admission. But these studies all share a common problem; they were conducted in highly selective programs in which all admitted and enrolled students have already been screened for the kind of reasoning skills that are assessed by the GRE and undergraduate grades. The only reasonable conclusion from these studies is that among students with strong reasoning skills other factors will determine who graduates or is a highly productive researcher; these studies cannot make inferences concerning the likely success of students with less developed reasoning abilities. The study that concluded that "GRE scores were significantly higher for men who left than counterparts who completed STEM PhD degrees" [3] was conducted in four flagship state universities. The problem with this conclusion is that virtually no one in these selective universities had low GRE-Q scores. The men who left had average scores of 742 and the completers had average scores of 723 (or about the 65th percentile on the old 200–800 GRE scale). So what does this tell us about the likely success of students with low or mediocre GRE scores, or about the potential value of the GRE?—absolutely nothing. The study entitled, "The limitations of the GRE in predicting success in biomedical graduate school" [2] relied on data from the highly selective biomedical graduate program at Vanderbilt University Medical School. While the authors correctly noted the lack of a correlation between GRE scores and publication count, none of the students in their sample had low GRE scores. Indeed, none of the students with three or more publications had GRE-Q scores below 550 and half had scores of at least 700. So, while the conclusion of a near zero correlation is correct, it is also true, and arguably much more relevant, that none of the students with three or more publications had low GRE-Q scores. Similarly, another study of first author publications was conducted in the highly selective biomedical program at the University of North Carolina-Chapel Hill [7]. Once again, it was true that GRE-Q scores are uncorrelated with first author publications. But a closer look reveals that the data can tell a different story. All of the students in this select program had high GRE scores; of the students with 3 or more first author publications, 84% had GRE scores at the 60th percentile or higher, and half scored at the 80th

percentile or higher. This leads to the clear conclusion that students with a strong record of first author publications tend to have high GRE-Q scores.

Another recent study [8] had the very provocative title, "Typical physics Ph.D. admissions criteria limit access to underrepresented groups but fail to predict doctoral completion," but the actual text indicated that completion is difficult to predict from either test scores or undergraduate grades. Nevertheless, they concluded that significant associations exist. Using a multivariate logistic model with the 3,692 physics students in their sample, they noted in their abstract, "Significant associations with completion were found for undergraduate GPA in all models and for GRE Quantitative in two of four studies models." The GRE-Q result is likely a substantial underestimate of the actual predictive power of the GRE because of a number of technical issues in the analysis [9]. Specifically, the Abstract of this critique noted, "The paper makes numerous elementary statistics errors, including introduction of unnecessary collider-like stratification bias, variance inflation by collinearity and range restriction, omission of needed data (some subsequently provided), a peculiar choice of null hypothesis on subgroups, blurring the distinction between failure to reject a null and accepting a null, and an extraordinary procedure for radically inflating confidence intervals in a figure."

It is almost impossible to find studies predicting valued outcomes in programs that have admitted students with low GRE scores and/or other indicators of less developed reasoning skills. Sealy, Saunders, Blume, and Chalkley [10] acknowledged "the typical biases of most GRE investigations of performance where primarily high-achievers on the GRE were admitted" (Abstract). They indicate a further limitation of many of the studies (including their own) with essentially null results relating GRE scores to publications or first author publications. Specifically, they note, "We are well aware that counting papers, either first author or total, has limitations–especially since neither metric captures the quality and/or impact of the publications" (p. 9). Their study followed a small (32 student) cohort of students who were carefully screened for admissions by a number of relevant criteria that did not include GRE scores. Statistical tests of relationships with GRE scores are not very meaningful in such a small sample, but it is worth noting that 28 of the 32 students in the program obtained PhD degrees. With careful selection on multiple criteria and intensive mentoring after enrollment, it is certainly true that students with relatively low GRE scores can succeed. But these data can neither support nor refute the possible relevance of GRE scores as part of a holistic review process. That is, strong GRE scores from a candidate could still boost the admissions chances for a student who was slightly lower on some of the other admissions criteria, such as the quality of the undergraduate institution attended.

In addition to program completion, another valued outcome for graduate programs is writing skill. Strong writing skills are required in many graduate courses and in all doctoral programs with a thesis requirement. There is evidence that the GRE Analytical Writing test predicts graduate grades across a number of graduate programs. Indeed, in a comprehensive study using data from over 25,000 students from 10 universities in the Florida state system the GRE Analytical Writing (GRE-AW) test was a significant predictor of the graduate grade point average across a number of different programs [6]. GRE-AW was frequently a better predictor than either the GRE-V or GRE-Q scores, perhaps surprisingly predicting grades in master's engineering programs and biomedical PhD programs better than predictions from GRE-Q. Because many factors in addition to writing skill are important in determining the overall grade point average, this study could not provide a direct link between GRE-AW and writing demands in graduate courses.

## Study 1

Successful completion of a graduate program is highly valued and may be the most important criterion for validation of pre-admission scores [e.g., 2, 3]. But any research using this criterion is problematic because students drop out for many reasons that are unrelated to reasoning skills and could not reasonably be expected to be predicted by test scores or undergraduate grades. A survey of students who left graduate school was conducted by researchers at the National Center for Education Statistics [11]. The survey indicated that the top eight reasons for leaving were: change in family status, conflict with job or military, dissatisfied with program, needed to work, personal problems, other financial reasons, taking time off, and other career interests; note that lack of necessary reasoning skills is not on this list. Nevertheless, if test scores are to be used as part of an admissions decision, it is reasonable to investigate whether there is any relationship of scores to program completion.

## Materials and methods

We had requested data from graduate programs representing a variety of selectivity levels but were ultimately successful in obtaining data from only two universities. GRE scores and program completion data were obtained from four highly selective PhD programs at a large flagship state university and at a highly selective Ivy League university. We understood that finding significant relationships to dropout in highly selective programs would likely be challenging, but even in these selective programs there was some variation in GRE scores, albeit near the top of the score scales. We intended to look at large programs in the social sciences and STEM. Specifically, we targeted programs in Chemistry, Electrical and Computer Engineering (ECE), History, and Psychology. The History and Psychology programs were eliminated from the analyses because only a handful of students in these programs left without a degree. We computed the correlation of GRE scores to dropout (0–1), but because the practical significance of correlations is frequently misunderstood [12, 13], we focused on more intuitive quartile comparisons that contrast the percentage of students with bottom quartile GRE scores who dropped out or stayed compared to the percentage of students with top quartile scores who dropped out or stayed. We omit the two middle quartiles to simplify the tables and focus attention on the contrast of high-scoring and low-scoring examinees. Quartiles were defined within programs within universities, so the bottom quartile in the Chemistry programs is not necessarily the same as the bottom quartile in the Electrical and Computer Engineering programs. For the fields in which there were substantial numbers of students who left without their intended degree (Chemistry and ECE), we noted the number of students at different GRE score levels (25th and 75th percentiles) who left before obtaining the degree and computed logistic regressions predicting the 0–1 outcome of dropout or stayed from the three GRE scores (Verbal [GRE-V], GRE-Q, and GRE-AW). We labelled the students who had not dropped out as "stayed," but note that at the time of the retrospective data collection most of the students who had not dropped out had already attained their PhD degrees, although a few were still enrolled. Students who enrolled in the PhD program but exited with a master's degree are counted as dropouts from the PhD program. None of the analyses for the GRE-V and GRE-AW scores indicated any significant differences by enrollment status; that is, in these samples GRE-V and GRE-AW were not significant predictors of program completion. Therefore, we focused primarily on the GRE-Q scores. This study was reviewed and approved by the ETS IRB (Committee for the Prior Review of Research; FWA00003247). The data file contained no personally identifiable information so individual consent was not required.

**Table 1. Chemistry dropouts By GRE-Q quartile.**

| GRE-Q Quartile | Stayed | Dropped Out | Total | % dropped |
|---|---|---|---|---|
| HiQ (75th %ile and above) | 68 | 11 | 79 | 14 |
| LoQ (25th %ile and below) | 55 | 24 | 79 | 30 |

## Results for chemistry programs

The Chemistry programs in both universities were highly selective with mean GRE-Q scores of 160 (SD = 5.7) for the 117 students in the flagship state university and 163 (SD = 5.8) for the 198 students in the Ivy League university. The 25th and 75th percentiles were 157 and 164 respectively at one university, and 158 and 167 at the other. In the tables we refer to the 25th percentile scores as ("LoQ") but recall that although these are relatively low scores in these highly selective universities, they are still well above the average for all examinees who took the GRE. (Among all GRE test takers: Mean = 154, SD = 9.5 [14]). Table 1 presents the Chemistry PhD dropouts by GRE-Q quartile.

Although the correlation of GRE-Q scores with 0–1 dropout was "only" -0.18, there were twice as many dropouts in the low GRE-Q group compared to the high GRE-Q group. This practically significant difference is also statistically significant. Chi-square, with Yates correction for 2x2 tables, is 5.28, $p < .03$.

Although the quartile comparison is dramatic and easily understood, it does not use all of the data. The maximum likelihood estimates from the logistic regression using all GRE scores are in Table 2. Similar to an ordinary least squares regression, the logistic regression provides an estimate of the importance and statistical significance of each predictor in the equation but is appropriate when the criterion is dichotomous (0–1 dropout or stay).

The Wald chi-square tests the statistical significance of the three GRE scores in the prediction model. Again, the GRE-Q is shown to be a significant predictor of dropout ($p < .002$). Note that the negative sign in the table is because dropout was coded as 1 and stay was coded as 0, so a negative sign indicates that students with low GRE-Q scores were more likely to drop out.

## Results for Electrical and Computer Engineering (ECE) programs

The ECE programs in both universities were highly selective with mean GRE-Q scores of 166 (SD = 4.0) for the 233 students in the flagship state university and 166 (SD = 4.1) for the 156 students in the Ivy League university. The mean scores at both universities were within 4 points of the maximum (170). The 25th and 75th percentiles were 164 and 170 respectively at both universities. Note that because of ties, the 75th percentile could also be the highest score possible. Table 3 presents the ECE PhD dropouts by GRE-Q quartile.

**Table 2. Maximum likelihood estimates for dropout from chemistry programs (N = 315).**

| Analysis of Maximum Likelihood Estimates | | | | | |
|---|---|---|---|---|---|
| Parameter | DF | Estimate | SE | Wald Chi-Square | Pr > ChiSq |
| Intercept | 1 | 5.6234 | 4.7297 | 1.4136 | 0.2345 |
| GREVerbal | 1 | 0.0345 | 0.0267 | 1.6678 | 0.1966 |
| GREQuantitative | 1 | -0.0766 | 0.0246 | 9.6965 | 0.0018 |
| GREWriting | 1 | 0.013 | 0.2339 | 0.0031 | 0.9558 |

Note.—DF is degrees of freedom; SE is standard error; Pr>ChiSq indicates statistical significance of the Chi Square.

**Table 3. ECE dropouts by GRE-Q quartile.**

| GRE-Q Quartile | Stayed | Dropped Out | Total | % dropped |
|---|---|---|---|---|
| HiQ (75th %ile and above) | 74 | 23 | 97 | 24 |
| LoQ (25th %ile and below) | 61 | 36 | 97 | 37 |

**Table 4. Maximum likelihood estimates for drop out from ECE programs (N = 389).**

| Analysis of Maximum Likelihood Estimates | | | | | |
|---|---|---|---|---|---|
| Parameter | DF | Estimate | SE | Wald Chi-Square | Pr > ChiSq |
| Intercept | 1 | 3.4207 | 4.9918 | 0.4696 | 0.4932 |
| GREVerbal | 1 | 0.008 | 0.0211 | 0.1438 | 0.7045 |
| GREQuantitative | 1 | -0.0315 | 0.0274 | 1.3153 | 0.2514 |
| GREWriting | 1 | -0.0798 | 0.1993 | 0.1604 | 0.6888 |

There were more drop outs in the "low" GRE group, but this difference fell short of the conventional standard for statistical significance. (Chi-square with Yates correction = 3.51, $p$ = .06). As indicated in Table 4, including all GRE scores (not just the high and low extremes) in the logistic regressions presents an even weaker case for the utility of GRE scores in predicting drop out.

But what this table may actually demonstrate is the folly of trying to make predictions from scores that are clustered at the top of the scale. In one school, the median score was 168 (2 points from the top of the 130–170 scale), and at the other school the median was 166. At both schools, the 75th percentile score was the highest score possible (170). At both schools, the domestic population (U. S. citizens and permanent residents) had somewhat lower GRE-Q scores than the international population, but still had very high scores relative to the 154 average for all test takers. The mean score for domestic students was 164 and the mean for international students was 167. The standard deviation of about 4 in both schools was substantially below the 9.5 standard deviation in the total testing population [14]. As indicated in Table 5, within this *slightly* lower scoring domestic population a significant relationship of GRE-Q scores to dropout emerged in the logistic regression analyses ($p$<.03).

## Study 1 conclusions

Results from the chemistry programs clearly indicate that GRE-Q scores can be effective in identifying students with a higher likelihood to drop out. There were twice as many dropouts in the bottom GRE-Q quartile compared to students in the top quartile. Results for the ECE programs were more ambiguous given the near-ceiling GRE scores of many of these students. Although some correction for the restricted range of the predictor scores is possible in

**Table 5. Maximum likelihood estimates for domestic dropout from ECE programs (N = 110).**

| Analysis of Maximum Likelihood Estimates | | | | | |
|---|---|---|---|---|---|
| Parameter | DF | Estimate | SE | Wald Chi-Square | Pr > ChiSq |
| Intercept | 1 | 21.3634 | 9.6258 | 4.9257 | 0.0265 |
| GREVerbal | 1 | -0.0385 | 0.0423 | 0.8317 | 0.3618 |
| GREQuantitative | 1 | -0.1112 | 0.0493 | 5.0878 | 0.0241 |
| GREWriting | 1 | 0.5268 | 0.3525 | 2.2342 | 0.135 |

correlational studies (e.g., Kuncel et al. [5]), the correction depends on fitting a regression line based on data that can be problematic when the restriction is as severe as it was in this study. When the slightly lower (but still very high) scores of the domestic students were analyzed separately, the ability of GRE-Q scores to predict dropout was again demonstrated. These tendencies should not be confused with destiny as many students with relatively low scores completed their degrees just as many students with relatively high scores dropped out. The main difficulty with these analyses is that they cannot speak to the fate of students with average or below GRE scores as such students simply do not exist in this dataset.

## Study 2

The ability to write logically, clearly, and grammatically is essential in any graduate program and in career success after graduate school. Writing actually done as part of graduate coursework assignments then appears to be an important validity criterion for pre-admissions test scores.

## Materials and methods

An initial attempt to obtain data on actual student writing in graduate courses involved contacting students in 15 universities that represented different levels of selectivity from a broad geographical spectrum. This effort was only minimally successful, so a second strategy entailed direct e-mail contact to students who had taken the GRE. Students from over 100 different universities responded to this effort.

Study participants were asked to submit the two most recent examples of writing that they had done in their graduate courses. We asked that the submitted papers be word-processed and approximately ten pages or fewer in length. Participants were permitted to send essays, term papers, book reports, and proposals, for example, but not very brief documents such as poems or other papers that did not contain fairly extended discourse (e.g., papers that consisted primarily of equations). Finally, study participants were encouraged (but not required) to send samples of their writing in which they (a) considered various perspectives and viewpoints or (b) constructed or analyzed arguments. This study was reviewed and approved by the ETS IRB (Committee for the Prior Review of Research: FWA00003247). Written (e-mail) consent was obtained from all participants.

Although grades assigned by professors constitute a readily usable criterion for course-related samples, they are, in all likelihood, based on widely different standards for each professor. Therefore, the approach taken followed procedures used in a previous study of student writing in realistic classroom contexts, and the scoring approach also closely matched the procedures in the previous study [15]. Student writing samples were evaluated according to a common set of criteria that could be used across submissions from the diverse set of graduate institutions. The criteria that we applied are those that were developed by four external experts in writing instruction/assessment. These criteria incorporate some of the scoring criteria on the GRE Analytical Writing measure for the "Issue" and "Argument" prompts. The Issue prompts ask the examinee to create an argument while the Argument prompts ask the examinee to critique an argument presented in the prompt. For the current study these holistic guides were expanded in order to reflect a concept of critical thinking that was characterized by one of the experts as being indicative of "scholarly habits of mind."

The writing was scored on a six-point rubric. The description for a "strong" essay (score of 5 [1 point below the top score of 6]) on this rubric is:

A 5 paper displays a generally thoughtful, well-developed treatment of the subject/topic and demonstrates strong control of the elements of writing.

A typical paper in this category

- discusses ideas or phenomena in some depth through analysis, synthesis, and/or persuasive reasoning

- develops and supports main points with logical reasons, examples, and/or details

- provides a generally well-focused, well-organized presentation, connecting ideas with clear transitions

- expresses ideas and information clearly, using language and varied sentence structure appropriate for the paper's context and content

- demonstrates facility with the conventions (i.e., grammar, usage, and mechanics) of standard written English but may have occasional flaws

Twelve college and university faculty members (all teachers and/or experienced evaluators of writing) evaluated all of the writing samples that were submitted. They were trained with a benchmark set of exemplary papers to represent each score level and a rangefinder set that spanned various disciplines within each set.

Of the course-related writing samples submitted by study participants, a total of 434 (two from each participant) were deemed to be scorable for the purposes of the study. As a collection, the samples were extremely varied with respect to numerous dimensions, including but not limited to (a) length, (b) content, (c) purpose, and (d) the conditions under which they were written. Submissions included literature reviews, critical analyses, mid-term essays, take-home examinations, critiques/evaluations, biographies, summaries, and so forth. Some appeared to be the result of semester-long efforts, while others seemed to be only one of numerous similar assignments required during an academic period. The wide variety of content found in the submissions is perhaps best illustrated simply by mentioning a few titles:

- "Abundance and sensitivities of the Eastern mud snail, Ilyansassa obsoleta, throughout the intertidal zone in Charleston, South Carolina"

- "Public relations: More than just press releases"

- "Living with severe mental illness: What families and friends must know: Evaluation of a one-day psychoeducation workshop"

- "The cognitive etiology of body dysphoric disorder (excessive concern over one's perceived physical flaws)"

With regard to length, submissions included:

- a 225 word application to participate in a summer math/statistics workshop

- a 2600 word essay on "How Netflix's business model changed to meet the demands of the consumer"

- a 5,000 word "numerical investigation" entitled "Optimization of a tandem blade configuration in an axial compressor," and

- a 6,000 word "sociological perspective" on the People's Republic of Korea entitled "Can North Koreans speak?"

**Table 6. Correlation of GRE scores with scores from course-required writing samples.**

| Scores | Mean | SD | Correlation with Class Writing Samples |
|---|---|---|---|
| GRE-V | 155 | 8 | 0.37* |
| GRE-Q | 152 | 8 | 0.04 |
| GRE-AW | 4.2 | 0.7 | 0.35* |
| Course Writing | 4 | 0.9 | - - |

Note. N = 217;

* $p < .00001$

In order to establish the rater reliability for the course-required writing tasks, 214 of the 434 writing tasks submitted were randomly selected to be read independently by two readers. For each essay, the score from the first rater (randomly selected from the pool of 12 raters) was correlated with the score from the second randomly selected rater. The inter-reader correlation for a single essay was .70. The score used as the criterion was the average of the scores on both writing tasks, which had a task reliability estimate of .54. This estimate is conservative because there is no expectation that the two writing tasks submitted should be strictly parallel.

## Results and discussion

Correlations of the scores on the class writing samples with GRE scores are presented in Table 6.

Because of the diversity in GRE scores of the students submitting writing samples, the standard deviations in the sample were only slightly lower than the standard deviations in the population (9, 9 and 0.9 for V, Q, and AW respectively [14]), so corrections for range restriction were not necessary. Not surprisingly, GRE-Q scores were not correlated with this writing criterion.

An alternative way of looking at the same data, as shown in Table 7 (and as we did in Study 1), is to note the percent of students who are low on the predictor and high on the criterion, or vice-versa.

Among students with low GRE-AW scores, there are seven times as many students with low scores on the criterion compared to students with high scores on the criterion. And among students with high GRE scores there are more than three times as many students with high criterion scores compared to those with low criterion scores. Chi-square, with Yates correction, is 19.2, $p < .00001$. These analyses demonstrate that GRE-AW scores are both statistically and practically significant indicators of writing skills in actual samples from graduate courses.

**Table 7. Percent of students with relatively high or low scores on course-required writing by GRE-AW score categories.**

| Course-required Writing | GRE-AW Low (3.5 and below) | GRE-AW High (5.0 and above) |
|---|---|---|
| High (5.0 and above) | 4 (4%) | 14 (18%) |
| Low (3.5 and below) | 29 (29% | 4 (5%) |
| Total *n* | 99 | 80 |

Note.—To simplify the table, students scoring between 3.5 and 5.0 were omitted.

## Conclusion

The two studies summarized here clearly demonstrate that even in highly selective programs GRE scores can indeed predict meaningful criteria that go beyond graduate GPA. Additional research is needed to better understand the role of GRE scores in less selective programs, and to evaluate possible differential effects in racial/ethnic and gender groups that could not be evaluated in this study because of the limited sample sizes for these groups. More elaborated regression models, or random tree models, that account for additional predictors or covariates such as undergraduate grades or socioeconomic statue should also be considered but note that such variables are often difficult or impossible to interpret in populations with large numbers of international students with undergradute grades on different scales and with socioeconomic indicators that may have different meanings internationally. Future research should provide more detailed analyses related to the prediction of writing skills. Specifically, the match between the rater's area of expertise and the assigned writing task should be explored, and with a larger sample analyses within specific program areas should be feasible.

But predicting dropout or writing ability in meaningful classroom contexts is only part of the story. A critic of tests such as the GRE could argue that placing too much emphasis on a single predictor could be detrimental to enrolling a diverse class. *And we agree*. But overreliance on a test score should not be confused with totally ignoring test scores. Without test scores, too much emphasis might be placed on other criteria, especially the undergraduate institution attended. Although accepting students from only top-ranked universities may help with enrolling a qualified class, it would be detrimental to enrolling a diverse class. This is because of a well-known problem with undermatch for many students from underrepresented minority groups. That is, because of financial or family considerations many minority students do not attend selective undergraduate schools for which they are fully qualified, as indexed by SAT scores or high school grades [16]. Institutions that dropped test scores and focused primarily on the undergraduate institution attended would miss these students. For these students, a GRE score could be an important opportunity, and possibly the only opportunity, to convincingly demonstrate their readiness for graduate school. With ample evidence available on the value of enrolling a diverse array of students [17], it would be unfortunate to ignore any measures that could help with this effort. While it is certainly wise to guard against too much reliance on test scores, it would be unwise to ignore scores that are related to meaningful indicators of success in graduate school and that may be the only way for some students to open the door to a graduate education.

## Acknowledgments

For Study 1, thanks to Duanli Yan and Wenju Cui for data analysis assistance, and special thanks to the graduate deans and staff at the participating universities for providing the data. For Study 2, thanks to Donald Powers for consultation on the design. We are particularly grateful to the graduate students who shared their written products with us.

## Author Contributions

**Conceptualization:** Brent Bridgeman, Frederick Cline.

**Formal analysis:** Frederick Cline.

**Methodology:** Brent Bridgeman.

**Writing – original draft:** Brent Bridgeman.

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
