## [Decision Letter · Decision Letter 0]

23 Nov 2021

PONE-D-21-06540

Does the GRE General Test predict more than just first year graduate GPA?

PLOS ONE

Dear Dr. Bridgeman,

Thank you for submitting your manuscript to PLOS ONE. After careful consideration, we feel that it has merit but does not fully meet PLOS ONE’s publication criteria as it currently stands. Therefore, we invite you to submit a revised version of the manuscript that addresses the points raised during the review process.

Although the two reviewers acknowledged the merits of this paper, it still needs improvement in various aspects, such as establishing reliability, strengthening argument and explicating the context of each study.

We look forward to receiving your revised manuscript.

Kind regards,

Mingming Zhou, Ph.D.

Academic Editor

PLOS ONE

Journal Requirements:

2. Please consider changing the title so as to meet our title format requirement (https://journals.plos.org/plosone/s/submission-guidelines). In particular, the title should be "Specific, descriptive, concise, and comprehensible to readers outside the field" and in this case it is not informative and specific about your study's scope, methodology, or its findings.

3. Thank you for including your ethics statement:  "Ethics statement for study 1:

This study was reviewed and approved by the ETS IRB (FWA00003247). The data file contained no personally identifiable information so individual consent was not required.

For study 2:This study was reviewed and approved by the ETS IRB (FWA00003247). Written (e-mail) consent was obtained from all participants.".   

"I have read the journal's policy and the authors of this manuscript have the following competing interests:  Financial support for this research was provided by Educational Testing Service which encourages the independence of researchers; nothing in this report represents an official position of ETS."" 

Reviewers' comments:

Reviewer's Responses to Questions

**Comments to the Author**

1. Is the manuscript technically sound, and do the data support the conclusions?

Reviewer #1: Yes

Reviewer #2: Partly

2. Has the statistical analysis been performed appropriately and rigorously? 

Reviewer #1: Yes

Reviewer #2: Yes

3. Have the authors made all data underlying the findings in their manuscript fully available?

Reviewer #1: Yes

Reviewer #2: Yes

4. Is the manuscript presented in an intelligible fashion and written in standard English?

Reviewer #1: Yes

Reviewer #2: Yes

5. Review Comments to the Author

Reviewer #1: Overall, this is a clear, concise, and relevant report presenting two studies. The first explored the utility of the GRE in predicting drop out of students in highly selective graduate programs in chemistry and computer engineering. The second explored the relationship between student GRE scores and writing ability in graduate school. In my opinion, this is a valuable contribution to the literature. I recommend several modest clarifications/additions below.

p. 2 – Last Paragraph and the first paragraph of page 3 discuss the difference between bias and demonstration of group differences. This discussion is both valuable and relevant. Of course, this does not establish that bias is not present in the GRE, only that observed group differences are not necessarily owing to bias. This discussion would be strengthened if the authors were able to briefly discuss the way that the development of the GRE minimizes the possibility of bias and/or share the results of studies establishing that group differences in GRE scores are not likely owing to bias.

p. 11 -- Study 1 Conclusions discuss the fact that these analyses “cannot speak to the fate of students with average or below GRE scores” because such students don’t exist in the dataset. While you did not use this analytic approach, it may be worth mentioning in the discussion that others such as Donald E. Powers 2004 paper in Journal of Applied Psychology: “Validity of Graduate Record Examinations (GRE) General Test Scores for Admissions to Colleges of Veterinary Medicine” have seen effects of range restriction for GRE scores.

p. 15. – I believe that a little more detail would be helpful to describe the process for establishing rater reliability. It appears that 214 of the 434 writing tasks were read by two readers (presumably 2 who were randomly paired from the pool of 12, and, I imagine, a separately randomly-assigned pair in each case, but that needs to be clarified, please. Also, while perhaps it should be, it is not clear to me how the inter-rater correlation was derived. (0.70). Please clarify.

Conclusion: The introduction and conclusion of the paper both discuss one of the current primary criticisms of the GRE – that it may show bias in favor of men and certain racial groups. The paper addresses those criticisms by providing validity evidence for the GRE, and arguing that if the GRE is abandoned, schools will be left to rely on fewer admissions criteria (such as selectivity of the undergraduate institution) which are, themselves likely to produce biased admissions results. I personally find this argument compelling. However, the paper does not directly compare/report on potential (or evident) bias in the data it examines. I understand that, if group differences were discovered, it would be difficult or impossible to establish whether they were owing to bias or existing sub-group differences. Nonetheless, I would be interested in the authors discussing why they chose not to compare group differences, given the purpose of the paper.

Reviewer #2: ---

The manuscript “Does the GRE General Test predict more than just first year graduate GPA” analyzes the extent to which different components of the GRE are predictive of PhD completion (Study 1) and future quality of graduate writing (Study 2).

Study 1 utilizes GRE score and completion data from 4 PhD programs (History, Psych, Chemistry, and ECE) at 2 institutions (a large state school and an Ivy league university). However, only the data from Chemistry and ECE are analyzed.

The authors first split the GRE scores into quartiles, and removed the middle two in order to create two groups of GRE scorers, “high” and “low.” Subsequent Chi-square tests revealed a statistically significant association between GRE-Q score category (“low” or “high”) and PhD completion for Chemistry majors but not ECE majors. The authors also perform a logistic regression using GRE-V, GRE-Q, and GRE-AW as independent variables and PhD completion as the dependent variable. Again, GRE-Q is found to be statistically significant for Chemistry majors but not ECE majors. However, among US ECE majors only, a significant relationship exists between GRE-Q and completion.

In all of Study 1’s analyses, GRE-V and GRE-AW are not significantly associated with completion at the 0.05 level.

Study 2 utilizes writing samples from students “from over 100 different universities” to analyze whether the GRE is associated with quality of graduate writing. Four “external experts in writing instruction/assessment” generated a common set of criteria by which to grade the samples. 12 college and university faculty then evaluated the writing samples on a six-point rubric. Correlations between the scored samples and the students’ GRE-V scores, as well as the students’ GRE-AW scores, were statistically significant.

From Study 1 and Study 2, the authors conclude that GRE scores can “predict meaningful criteria that go beyond graduate GPA.”

---

The issues surrounding the use of GRE scores in graduate admissions processes are both topical and extremely important to the academic community, making this manuscript a valuable contribution to the ongoing discourse in the literature. Indeed, as noted in the manuscript, studies predicting valued outcomes in PhD programs admitted with a broad array of GRE scores are rare. The data presented in Study 2 is especially unique and should provide a tremendous starting point for similar studies in the future. The statistical analyses performed are reasonable for the data collected and are executed well.

However, I believe the manuscript would be strengthened by the consideration of several edits/additions by the authors. Notably, the issues outlined in the “Overall Conclusions” section are the primary reason for indicating that the data “Partly” support the conclusions.

Literature review -

1) The authors appropriately note that studies of GRE scores often do not address issues related to range restriction. However, they use phrases like “low” and “mediocre” GRE scores to describe the ranges of students “missing” from these datasets, but provide no context for what is actually considered “low.” For instance, noting that the average GRE-Q score of completers in [3] was 723 means little without knowing where that sits in context (a quick search of ETS indicates that’s around a 65th percentile, which some faculty likely would not consider a “high” GRE-Q score). The issue of keeping track of which studies have larger ranges of scores than others is further muddled by the fact that the authors use the new scoring scale in their analysis, but use the old scale in the literature review. How does the data in this manuscript compare to previous studies, which the manuscript describes as saying “absolutely nothing” about the value of the GRE? If anything, a limited range in Study 1 would actually serve to bolster the correlation of GRE-Q and completion that the authors find, but I think that point is lost. Perhaps a table summarizing all of the data, with percentiles and old/new score conversions, would be effective.

2) A reference on range restriction (e.g. https://arxiv.org/abs/1709.02895) might help readers unfamiliar with these issues.

3) Authors should consider citing Kuncel et al. (2001), which actually corrects for range restriction issues and is often cited on the ETS website, but finds no significant correlation between GRE-Q (or -V) and PhD completion in their meta-analysis. How do the authors reconcile this result with their current manuscript?

4) A follow-up to Ref [8] (https://doi.org/10.1103/PhysRevPhysEducRes.17.020115), in general, finds the same basic conclusions as this manuscript (that GRE-Q is correlated with PhD completion), and addresses several concerns laid out in [9]. As such it is likely worth citing.

5) Lastly, the literature review is entirely dedicated to studies most related to Study 1, in particular the existing literature surrounding GRE-Q. Something needs to be said about Study 2 in order to frame it within a larger context. Has there been previous work on standardized writing (on SATs? APs?) and success?

Study 1 -

6) The authors must either discuss in more detail the data collected on the History and Psychology programs or entirely remove the comment referring to their elimination from the study. Presumably the History majors had the largest spread of GRE-Q scores (as a non-STEM field), yet they all graduated anyway? Was this just a small N problem or are we seeing essentially no correlation between GRE and completion within history majors (since they’re all graduating independent of score)? This deserves further comment.

7) While it is understandable to want “intuitive” results, some readers will immediately note that by splitting data into quartiles, the authors are “throwing away” some information in the data. Consider including both the correlation and 2x2 tables even if some readers may "misinterpret" correlations.

8) With regard to interpretability, the audience of faculty on admissions committees for whom this paper would most likely be targeted may be unfamiliar with logistic regression using independent variables that have been split into quartiles. A brief comment on what the estimates in Tables 2/4/5 mean in terms of practical score changes and probability of completion would be helpful.

Study 2 -

9) As mentioned in comment 5), Study 2 lacks the same context as Study 1. As it stands, essentially the entire discussion in the introduction is dedicated to the research landscape as it pertains to GRE-Q, with little to no mention of GRE-V or writing. Hence Study 2 feels only loosely connected to the rest of the paper

10) In the same spirit as the previous comment, Study 2’s methods and data should be discussed in more detail. For example,

- How many students were included? What are the demographics of the students who submitted writing samples? What are their majors? What can the study say about majors not included?

- What are the demographics of the faculty who graded the responses? Do these align, and does it matter?

- For instance, the rubric appears to be constructed so that faculty from any discipline can grade papers from any other discipline (ie, a psychology faculty member might grade a history paper and evaluate its issue and argument) - is this justified? Can a faculty member determine whether an argument is good when it is out of disciplinary context? Perhaps, but this decision is not discussed. What information was lost by stripping disciplinary context?

- One possible avenue to explore this is to examine the scores given by the students’ instructors in comparison to those given by the independent graders.

- Have previous studies used a similar methodology to evaluate written work?

- What do the distributions of scores look like on the graded papers? What about the original grades? What do these say about the sample of students?

- Were there specific requirements for a paper to be “deemed scorable” by the scorers?

This list is surely non-exhaustive, and I believe there is much room for the authors to elaborate on how this very interesting data was collected and analyzed.

Overall conclusions -

This manuscript presents two studies supporting two separate but related claims (as written in the abstract): one is that this manuscript “demonstrate(s) the ability of the GRE to predict dropout” and the other is that this manuscript “shows the relationship of the GRE Analytical Writing scores to writing produced as part of graduate school coursework.”

The first claim, that “the GRE” predicts dropout, is only partly supported by the data, and lacks important context. First, as made clear by the manuscript’s focus on the GRE-Q, the authors only find evidence that the quantitative part of the exam correlates with completion - little can be said about the GRE-V. Since students have to take (and pay for) both, the authors should distinguish which parts of the exam they found to be significantly associated with completion, and be consistent about this throughout the manuscript. Amending the abstract to include details of the study (e.g. 4 programs studied at 2 universities, N = students, GRE-Q found significantly associated with completion for one of the programs) avoids any possible obfuscation of the findings.

More importantly however, the authors should clarify the implications of their findings. As noted earlier, several other papers have found a correlation between GRE-Q and PhD completion in a STEM discipline. But does finding a correlation between GRE-Q and completion automatically justify its use in admissions? It is in this area where the authors must elaborate. This manuscript notes that the most frequently cited reasons for dropping out of graduate school are known to be completely non-academic: financial issues, family, etc. It also notes that different groups are known to score differently on the GRE exam. Could it be that the GRE is simply picking up on some socioeconomic conditions related to, as the authors say “fewer educational opportunities and poorer schools,” which then make it more difficult to persist in graduate school when tempted with higher paying jobs or confronted with family emergency? Or is the GRE really picking up on reasoning skills that faculty value and help students persist? The authors provide no evidence that this is indeed the case, and actually provide a more compelling argument for the former interpretation. Put another way, why should faculty believe that a students’ ability to do high school math questions make them more able to complete a PhD? Or are other variables confounding this relationship? Does it matter, so long as the test is indicating whether students will drop out? I suspect critics of the GRE would wonder why admissions committees should force students to pay to take a test that largely measures the socioeconomic status of the test takers.

Lastly (again related to the fact that Study 2 is largely ignored throughout the introduction), the authors must reconcile statements such as “Although there is ample evidence that the scores can predict graduate school readiness as indexed by grades in the first one or two years (e.g., [5,6]) mere grade prediction is of limited value.” with the fact that Study 2 uses writing samples produced in graduate coursework. Is that not merely a prediction of graduate grades? Does this really link to valued outcomes such as “research ability” and “completion”? The authors tenuously make this link in the first sentence of Study 2, but they must make this link clearer in order for Study 2 to be used in support of the paper’s overall conclusion that the GRE predicts more than grades.

6. PLOS authors have the option to publish the peer review history of their article (what does this mean?). If published, this will include your full peer review and any attached files.

Reviewer #1: **Yes: **Jared A. Danielson

Reviewer #2: No

---

## [Author Response · Author response to Decision Letter 0]

25 Jan 2022

Response to reviewers

Reviewer 1

p. 2 –…This discussion would be strengthened if the authors were able to briefly discuss the way that the development of the GRE minimizes the possibility of bias and/or share the results of studies establishing that group differences in GRE scores are not likely owing to bias.

Added: “Trained fairness reviewers, including representatives of minority groups, review all GRE questions and statistical differential item functioning (DIF) procedures are used to identify any test questions that are unusually difficult (or easy) for a particular racial/ethnic or gender group.” 

p. 11 -- …While you did not use this analytic approach, it may be worth mentioning in the discussion that others such as Donald E. Powers 2004 paper in Journal of Applied Psychology: “Validity of Graduate Record Examinations (GRE) General Test Scores for Admissions to Colleges of Veterinary Medicine” have seen effects of range restriction for GRE scores.

I now mention range restriction in the context of the Kuncel et al. study that was mentioned in the introduction but note that the correction can be problematic when the restriction is severe. Although Powers (2004) is a good study it is not as relevant because the restriction on both predictors and criterion was modest in the sample of veterinary schools. 

Added: “Although some correction for the restricted range of the predictor scores is possible in correlational studies (e.g., Kuncel et al. [6]), the correction depends on fitting a regression line based on data that can be very sparse when the restriction is as severe as it was in this study.”

p. 15. – I believe that a little more detail would be helpful to describe the process for establishing rater reliability. It appears that 214 of the 434 writing tasks were read by two readers (presumably 2 who were randomly paired from the pool of 12, and, I imagine, a separately randomly-assigned pair in each case, but that needs to be clarified, please. Also, while perhaps it should be, it is not clear to me how the inter-rater correlation was derived. (0.70). Please clarify.

Added: “For each essay, the score from the first rater (randomly selected from the pool of 12 raters) was correlated with the score from the second randomly selected rater.”

Conclusion: … Nonetheless, I would be interested in the authors discussing why they chose not to compare group differences, given the purpose of the paper.

Sample sizes were too small to compare group differences. 

Added in the discussion: “Additional research is needed to better understand the role of GRE scores in less selective programs, and to evaluate possible differential effects in racial/ethnic and gender groups that could not be evaluated in this study because of the limited sample sizes for these groups.”

Reviewer 2

1) …For instance, noting that the average GRE-Q score of completers in [3] was 723 means little without knowing where that sits in context (a quick search of ETS indicates that’s around a 65th percentile, which some faculty likely would not consider a “high” GRE-Q score). 

Added that 723 is about 65th percentile. Also toned down comment on “high” scores on the GRE to say simply “none of the students with three or more publications had low GRE-Q scores.” 

2) A reference on range restriction (e.g. https://arxiv.org/abs/1709.02895) might help readers unfamiliar with these issues.

Referred reader back to the Kuncel study mentioned in the introduction in the context of range restriction.

3) Authors should consider citing Kuncel et al. (2001), which actually corrects for range restriction issues and is often cited on the ETS website, but finds no significant correlation between GRE-Q (or -V) and PhD completion in their meta-analysis. How do the authors reconcile this result with their current manuscript?

Kuncel was cited in the introduction, but it does not deserve too much attention. Many of the studies in the meta-analysis are over 30 years old when the test itself as well as the populations taking it were quite different than they are today. This why the current study was needed. 

4) A follow-up to Ref [8] (https://doi.org/10.1103/PhysRevPhysEducRes.17.020115), in general, finds the same basic conclusions as this manuscript (that GRE-Q is correlated with PhD completion), and addresses several concerns laid out in [9]. As such it is likely worth citing.

The follow-up states, “Students’ undergraduate GPA (UGPA) and GRE Physics (GRE-P) scores are small but statistically significant predictors of graduate course grades, while GRE quantitative and GRE verbal scores are not,” but corrects only a few of the problems outlined in [9], so it does not seem to be worth citing.

5) Lastly, the literature review is entirely dedicated to studies most related to Study 1, in particular the existing literature surrounding GRE-Q. Something needs to be said about Study 2 in order to frame it within a larger context. Has there been previous work on standardized writing (on SATs? APs?) and success?

Added: “In addition to program completion, another valued outcome for graduate programs is writing skill. Strong writing skills are required in many graduate courses and in all doctoral programs with a thesis requirement. There is evidence that the GRE Analytical Writing test predicts graduate grades across a number of graduate programs. Indeed, in a comprehensive study using data from over 25,000 students from 10 universities in the Florida state system the GRE Analytical Writing (GRE-AW) test was a significant predictor of the graduate grade point average across a number of different programs [6]. GRE-AW was frequently a better predictor than either the GRE-V or GRE-Q scores, perhaps surprisingly predicting grades in master’s engineering programs and biomedical PhD programs better than predictions from GRE-Q. Because many factors in addition to writing skill are important in determining the overall grade point average, this study could not provide a direct link between GRE-AW and writing demands in graduate courses.”

6) The authors must either discuss in more detail the data collected on the History and Psychology programs or entirely remove the comment referring to their elimination from the study. Presumably the History majors had the largest spread of GRE-Q scores (as a non-STEM field), yet they all graduated anyway? Was this just a small N problem or are we seeing essentially no correlation between GRE and completion within history majors (since they’re all graduating independent of score)? This deserves further comment.

Because our plan was to analyze results in four programs, dropping mention of the history and psychology does not seem to be appropriate. It is not exactly a small N problem but is a problem because of the very small N of students who drop out of these programs. It is not possible to predict dropout when students are not dropping out.

7) While it is understandable to want “intuitive” results, some readers will immediately note that by splitting data into quartiles, the authors are “throwing away” some information in the data. Consider including both the correlation and 2x2 tables even if some readers may "misinterpret" correlations.

Added correlation.

8) With regard to interpretability, the audience of faculty on admissions committees for whom this paper would most likely be targeted may be unfamiliar with logistic regression using independent variables that have been split into quartiles. A brief comment on what the estimates in Tables 2/4/5 mean in terms of practical score changes and probability of completion would be helpful.

Note that the logistic regression used the original data before it had been split into quartiles.

Added: “Similar to an ordinary least squares regression, the logistic regression provides an estimate of the importance and statistical significance of each predictor in the equation but is appropriate when the criterion is dichotomous (0-1 drop out or stay).”

9) As mentioned in comment 5), Study 2 lacks the same context as Study 1. As it stands, essentially the entire discussion in the introduction is dedicated to the research landscape as it pertains to GRE-Q, with little to no mention of GRE-V or writing. Hence Study 2 feels only loosely connected to the rest of the paper.

See response to comment 5.

10) In the same spirit as the previous comment, Study 2’s methods and data should be discussed in more detail. For example,

- How many students were included? What are the demographics of the students who submitted writing samples? What are their majors? What can the study say about majors not included?

- What are the demographics of the faculty who graded the responses? Do these align, and does it matter?

- For instance, the rubric appears to be constructed so that faculty from any discipline can grade papers from any other discipline (ie, a psychology faculty member might grade a history paper and evaluate its issue and argument) - is this justified? Can a faculty member determine whether an argument is good when it is out of disciplinary context? Perhaps, but this decision is not discussed. What information was lost by stripping disciplinary context?

- One possible avenue to explore this is to examine the scores given by the students’ instructors in comparison to those given by the independent graders.

- Have previous studies used a similar methodology to evaluate written work?

- What do the distributions of scores look like on the graded papers? What about the original grades? What do these say about the sample of students?

- Were there specific requirements for a paper to be “deemed scorable” by the scorers?

Some of these questions have been addressed, but data was not available for many of them. The similarity of the approach taken here to previous research [15] was acknowledged. 

Added: “Therefore, the approach taken followed procedures used in a previous study of student writing in realistic classroom contexts, and the scoring approach also closely matched the procedures in the previous study [15].” 

The need for additional work in this area was also acknowledged.

Added: “Future research should provide more detailed analyses related to the prediction of writing skills. Specifically, the match between the rater’s area of expertise and the assigned writing task should be explored, and with a larger sample analyses within specific program areas should be feasible.” 

… Amending the abstract to include details of the study (e.g. 4 programs studied at 2 universities, N = students, GRE-Q found significantly associated with completion for one of the programs) avoids any possible obfuscation of the findings.

Amended abstract: “…Two studies are included. The first study used data from chemistry (N=320) and computer engineering (N=389) programs from a flagship state university and an Ivy League university to demonstrate the ability of the GRE to predict dropout. Dropout prediction for the chemistry programs was both statistically and practically significant for the GRE quantitative (GRE-Q) scores, but not for the verbal or analytical writing scores. In the computer engineering programs, significant dropout prediction by GRE-Q was evident only for domestic students. The second study showed the relationship of GRE Analytical Writing scores to writing produced as part of graduate school coursework by 217 students. 

Put another way, why should faculty believe that a students’ ability to do high school math questions make them more able to complete a PhD?

Although GRE-Q does not require computational skills beyond high school math, the reasoning required is far beyond “ability to do high school math.” If only high school knowledge were required half of the examinees would not have scores below 154 and GRE-Q could not predict either grades or dropout.

Or are other variables confounding this relationship? Does it matter, so long as the test is indicating whether students will drop out?

There may be many intervening variables, but this does not alter the conclusion that GRE-Q scores are a useful indicator of students who are more likely to drop out.

…the authors must reconcile statements such as “Although there is ample evidence that the scores can predict graduate school readiness as indexed by grades in the first one or two years (e.g., [5,6]) mere grade prediction is of limited value.” with the fact that Study 2 uses writing samples produced in graduate coursework. Is that not merely a prediction of graduate grades?

No; writing is just one aspect of graduate grades and this study permits focus on just the writing skill component.

---

## [Decision Letter · Decision Letter 1]

31 Mar 2022

PONE-D-21-06540R1

Can the GRE Predict Valued Outcomes?  Dropout and Writing Skill

PLOS ONE

Dear Dr. Bridgeman,

Thank you for submitting your manuscript to PLOS ONE. After careful consideration, we feel that it has merit but does not fully meet PLOS ONE’s publication criteria as it currently stands. Therefore, we invite you to submit a revised version of the manuscript that addresses the points raised during the review process.

As you can read from the reviewers' comments, there are still some concerns to be solved. More importantly, the consistency between the response letter and the revised text needs to be ensured.

We look forward to receiving your revised manuscript.

Kind regards,

Mingming Zhou, Ph.D.

Academic Editor

PLOS ONE

Reviewers' comments:

Reviewer's Responses to Questions

**Comments to the Author**

1. If the authors have adequately addressed your comments raised in a previous round of review and you feel that this manuscript is now acceptable for publication, you may indicate that here to bypass the “Comments to the Author” section, enter your conflict of interest statement in the “Confidential to Editor” section, and submit your "Accept" recommendation.

Reviewer #1: All comments have been addressed

Reviewer #3: (No Response)

Reviewer #4: (No Response)

2. Is the manuscript technically sound, and do the data support the conclusions?

Reviewer #1: Yes

Reviewer #3: Partly

Reviewer #4: Yes

3. Has the statistical analysis been performed appropriately and rigorously? 

Reviewer #1: Yes

Reviewer #3: Yes

Reviewer #4: Yes

4. Have the authors made all data underlying the findings in their manuscript fully available?

Reviewer #1: Yes

Reviewer #3: Yes

Reviewer #4: Yes

5. Is the manuscript presented in an intelligible fashion and written in standard English?

Reviewer #1: Yes

Reviewer #3: Yes

Reviewer #4: Yes

6. Review Comments to the Author

Reviewer #1: Please note that the changes described in response to the reviewers are acceptable in my perspective. However, the change described in the author response to the reviewers regarding rater reliability does not appear in the new manuscript. I assume this is a simple oversight, and I assume it will be added.

Reviewer #3: This paper deals with the important issue of dropout from graduate programs and the ability to predict dropouts and graduates based on GRE scores. It adopts a differentiated approach, based on the different sub-sections of the GRE, which is a welcome contribution. The paper also presents the findings of not one but two studies. The first is a classic prediction one (focusing on GRE-Q) and the second present a novel approach to assess the worth of GRE-AW.

With that said, the paper also has some weaknesses. As I enter the review stage during the R&R, I think of the following comments as food for thought – maybe helpful for improving the introduction and conclusion.

Reading the paper, there is an unsettling dissonance. The authors criticize former studies for focusing on highly selective institutions and programs (with slight score variation). Yet, their study is based on data from highly selective institutions and openly states that history and psychology were eliminated because of minor score variations. It reads nearly as they are repeating the pitfalls they warn from. Relatedly, can the authors explain what is the source of this variation? One would think that in highly selective institutions, only top-performing students will be admitted. So what is unique in the admission criteria in these programs, that enabled this variation from the first place? One possible answer can be that in some fields there is greater willingness to admit “everyone,” starting with big classrooms and seeing who will pass the first-year exams. If this is the case, there is no point in the pre-admission tests from the first place.

Alternatively, “good” and “excellent” students were admitted. In such a case, the fact that the logistic regression includes only the GRE scores with no other background characteristics (undergraduate GPA, socioeconomic status, parental education and so forth) is puzzling. With more parameters within the models, they are very likely to show different results. Yet, we are left in the dark.

The analysis itself is ok. It would be good to explain why picking ML over other options for the logistic regression. And as there are a growing number of studies that turn to machine learning models that are ideal for prediction purposes (random forests is but one option), this statistical approach seems valid but outdated. The same applies to the sample size. I am well aware of the difficulty of collecting data on these issues, and the N is acceptable to achieve statistical significance, but still, it looks a bit low for today’s standards.

Turning to study 2:

Overall I found this part to be very interesting, and dealing with an issue that is less “objective” and harder to measure, which probably explains why there is little literature on the topic.

Is there a way to tell the readers at what stage in their programs the responders were when submitting the written examples? 20% in their first year, 50% in the second year, etc. It is likely to assume that over time, grad students will improve their writing skills, and the association with the GRE-AW will weaken.

I appreciate the quartile approach. Maybe a more detailed one would be also informative. I understand that you don’t have the data on ethnicity, gender, and so forth, but even just using a more fine-tuned classification will do. A good example is provided in Bowen and Bok book, the shape of the river (appendix table D.3.3).

I was surprised to see no reference to Bowen and Bok work. It is also possible to think about the connection between your argument (which is really somewhat of a “don't throw the baby out with the bathwater”) and the ongoing debates about affirmative action. I don’t think that your data can support direct confirmation to either side of the debate, but it would be beneficial to at least acknowledge it. When thinking about it, even the fact that you do not present any other variables in the regression models means something.

To conclude – address the dissonance I pointed out above, and explain the limitation and decision to focus solely on GRE in the regression. A bit of a conversation with the literature could help, too.

Reviewer #4: Page 1: Please, also include results of the second study in the abstract. For example: “Our analyses demonstrate that GRE-AW scores are both statistically and practically significant indicators of writing skills in actual samples from graduate courses.”

Page 5: [… GRE because of a number of technical issues in the analysis [9].] > List the technical issues briefly.

Page 8: Please, mention that, now, you also include correlation coefficients.

Page 10: The numbers of students included in the study on page 9 (117 from a flagship state university and 198 from an Ivy League university) sum up to 315 and not to 320 as indicated in the caption of table 2. Please check and correct the numbers if necessary.

Page 10: There is a space missing in the last paragraph: “were 164 and170”.

Page 12: In the result section you claim “Results from the chemistry programs clearly indicate that GRE-Q scores can be effective in identifying students most likely to drop out”. Given that you show that 30% of students with a low GRE-Q score drop out and that 14% with a high GRE-Q score drop out, I would rather suggest to claim that “… in identifying students with a higher likelihood to drop out”.

Page 21, reference 5: no space in “meta-analytic”.

Page 22, reference 12: “;” not in italic font & wrong dash in page numbers.

Page 22, reference 13: “Rubin, D” instead of “Rubin ,D” & “, 74; 1982,” not in italic font

7. PLOS authors have the option to publish the peer review history of their article (what does this mean?). If published, this will include your full peer review and any attached files.

Reviewer #1: **Yes: **Jared A. Danielson

Reviewer #3: No

Reviewer #4: No

---

## [Author Response · Author response to Decision Letter 1]

4 Apr 2022

Response to reviewers (April 2022)

the author response to the reviewers regarding rater reliability does not appear in the new manuscript

added: For each essay, the score from the first rater (randomly selected from the pool of 12 raters) was correlated with the score from the second randomly selected rater. The inter-reader correlation for a single essay was .70.

Reading the paper, there is an unsettling dissonance. The authors criticize former studies for focusing on highly selective institutions and programs (with slight score variation). ). Yet, their study is based on data from highly selective institutions

This was addressed by modifying the methods and materials for Study 1 as follows:

We had requested data from graduate programs representing a variety of selectivity levels but were ultimately successful in obtaining data from only two universities. GRE scores and program completion data were obtained from four highly selective PhD programs at a large flagship state university and at a highly selective Ivy League university. We understood that finding significant relationships to dropout in highly selective programs would likely be challenging, but even in these selective programs there was some variation in GRE scores, albeit near the top of the score scales. 

The analysis itself is ok. It would be good to explain why picking ML over other options for the logistic regression. And as there are a growing number of studies that turn to machine learning models that are ideal for prediction purposes (random forests is but one option), this statistical approach seems valid but outdated…. I understand that you don’t have the data on ethnicity, gender, and so forth, but even just using a more fine-tuned classification will do….explain the limitation and decision to focus solely on GRE in the regression

In the conclusion we acknowledged that other analytic approaches were possible, and by implication why we chose not to use some covariates (e.g., undergrad GPA) that might make sense in other contexts but were less relevant to the current context.

More elaborated regression models, or random tree models, that account for additional predictors or covariates such as undergraduate grades or socioeconomic statue should also be considered but note that such variables are often difficult or impossible to interpret in populations with large numbers of international students with undergraduate grades on different scales and with socioeconomic indicators that may have different meanings internationally.

I was surprised to see no reference to Bowen and Bok work. It is also possible to think about the connection between your argument (which is really somewhat of a “don't throw the baby out with the bathwater”) and the ongoing debates about affirmative action.

Bowen & Bok reference added in the Conclusion:

With ample evidence available on the value of enrolling a diverse array of students [17], it would be unfortunate to ignore any measures that could help with this effort.

Page 1: Please, also include results of the second study in the abstract

Added to abstract: . In the second study, GRE Analytical Writing scores for 217 students were related to writing produced as part of graduate school coursework and relationships were noted that were both practically and statistically significant.

Page 5: [… GRE because of a number of technical issues in the analysis [9].] > List the technical issues briefly.

Added: ]. Specifically, the Abstract of this critique noted, “The paper makes numerous elementary statistics errors, including introduction of unnecessary collider-like stratification bias, variance inflation by collinearity and range restriction, omission of needed data (some subsequently provided), a peculiar choice of null hypothesis on subgroups, blurring the distinction between failure to reject a null and accepting a null, and an extraordinary procedure for radically inflating confidence intervals in a figure.”

Page 8: Please, mention that, now, you also include correlation coefficients

Added: We computed the correlation of GRE scores to dropout (0-1),

Page 10: The numbers of students included in the study on page 9 (117 from a flagship state university and 198 from an Ivy League university) sum up to 315 and not to 320 as indicated in the caption of table 2. Please check and correct the numbers if necessary.

Fixed

Page 10: There is a space missing in the last paragraph: “were 164 and170”.

Fixed

Page 12: In the result section you claim “Results from the chemistry programs clearly indicate that GRE-Q scores can be effective in identifying students most likely to drop out”. Given that you show that 30% of students with a low GRE-Q score drop out and that 14% with a high GRE-Q score drop out, I would rather suggest to claim that “… in identifying students with a higher likelihood to drop out”.

Fixed: … in identifying students with a higher likelihood to drop out. 

Page 21, reference 5: no space in “meta-analytic”.

? I did not see any extra space. Spelling exactly as in the original reference.

Page 22, reference 12: “;” not in italic font & wrong dash in page numbers.

Page 22, reference 13: “Rubin, D” instead of “Rubin ,D” & “, 74; 1982,” not in italic font

fixed

---

## [Decision Letter · Decision Letter 2]

9 May 2022

Can the GRE Predict Valued Outcomes?  Dropout and Writing Skill

PONE-D-21-06540R2

Dear Dr. Bridgeman,

We’re pleased to inform you that your manuscript has been judged scientifically suitable for publication and will be formally accepted for publication once it meets all outstanding technical requirements.

Kind regards,

Mingming Zhou, Ph.D.

Section Editor

PLOS ONE

Additional Editor Comments (optional):

Reviewers' comments:

Reviewer's Responses to Questions

**Comments to the Author**

1. If the authors have adequately addressed your comments raised in a previous round of review and you feel that this manuscript is now acceptable for publication, you may indicate that here to bypass the “Comments to the Author” section, enter your conflict of interest statement in the “Confidential to Editor” section, and submit your "Accept" recommendation.

Reviewer #1: All comments have been addressed

Reviewer #3: All comments have been addressed

Reviewer #4: All comments have been addressed

2. Is the manuscript technically sound, and do the data support the conclusions?

Reviewer #1: Yes

Reviewer #3: Yes

Reviewer #4: Yes

3. Has the statistical analysis been performed appropriately and rigorously? 

Reviewer #1: Yes

Reviewer #3: Yes

Reviewer #4: Yes

4. Have the authors made all data underlying the findings in their manuscript fully available?

Reviewer #1: Yes

Reviewer #3: Yes

Reviewer #4: Yes

5. Is the manuscript presented in an intelligible fashion and written in standard English?

Reviewer #1: Yes

Reviewer #3: Yes

Reviewer #4: Yes

6. Review Comments to the Author

Reviewer #1: (No Response)

Reviewer #3: I thank the authors for the care with which they responded to my comments in previous round of revisions. In my opinion, this article is now ready for publication.

Minor clarification:

On page 7, “Because many factors in addition to writing skill are important in determining the overall grade point average, this study could not provide a direct link between GRE-AW and writing demands in graduate courses.” This study refers to citations No. 6? Please clarify

Reviewer #4: (No Response)

7. PLOS authors have the option to publish the peer review history of their article (what does this mean?). If published, this will include your full peer review and any attached files.

Reviewer #1: **Yes: **Jared A. Danielson

Reviewer #3: No

Reviewer #4: No

---

## [Editor Report · Acceptance letter]

12 May 2022

PONE-D-21-06540R2 

Can the GRE Predict Valued Outcomes?  Dropout and Writing Skill 

Dear Dr. Bridgeman:

I'm pleased to inform you that your manuscript has been deemed suitable for publication in PLOS ONE. Congratulations! Your manuscript is now with our production department. 

Kind regards, 

on behalf of

Dr. Mingming Zhou 

Section Editor

PLOS ONE